# Midazolam as a Probe for Heterotropic Drug-Drug Interactions Mediated by CYP3A4

**DOI:** 10.3390/biom12060853

**Published:** 2022-06-20

**Authors:** Ilia G. Denisov, Yelena V. Grinkova, Mark A. McLean, Tyler Camp, Stephen G. Sligar

**Affiliations:** 1Department of Biochemistry, University of Illinois at Urbana-Champaign, Urbana, IL 61801, USA; denisov@illinois.edu (I.G.D.); grinkova@illinois.edu (Y.V.G.); m-mclean@illinois.edu (M.A.M.); tylerc5@illinois.edu (T.C.); 2Center for Biophysics and Quantitative Biology, University of Illinois at Urbana-Champaign, Urbana, IL 61801, USA; 3Department of Chemistry, University of Illinois at Urbana-Champaign, Urbana, IL 61801, USA

**Keywords:** drug interactions, cytochrome P450 CYP3A4, midazolam, allosteric interactions, nanodiscs

## Abstract

Human cytochrome P450 CYP3A4 is involved in the processing of more than 35% of current pharmaceuticals and therefore is responsible for multiple drug-drug interactions (DDI). In order to develop a method for the detection and prediction of the possible involvement of new drug candidates in CYP3A4-mediated DDI, we evaluated the application of midazolam (MDZ) as a probe substrate. MDZ is hydroxylated by CYP3A4 in two positions: 1-hydroxy MDZ formed at lower substrate concentrations, and up to 35% of 4-hydroxy MDZ at high concentrations. The ratio of the formation rates of these two products (the site of metabolism ratio, SOM) was used as a measure of allosteric heterotropic interactions caused by effector molecules using CYP3A4 incorporated in lipid nanodiscs. The extent of the changes in the SOM in the presence of effectors is determined by chemical structure and is concentration-dependent. MD simulations of CYP3A4 in the lipid bilayer suggest that experimental results can be explained by the movement of the F-F’ loop and concomitant changes in the shape and volume of the substrate-binding pocket. As a result of PGS binding at the allosteric site, several residues directly contacting MDZ move away from the substrate molecule, enabling the repositioning of the latter for minor product formation.

## 1. Introduction

CYP3A4 is the main drug metabolizing cytochrome P450 in humans. The simultaneous binding of two or three substrate molecules results in non-Michaelis behavior because of homotropic or heterotropic interactions between substrates [1,2,3,4,5,6,7,8,9]. In addition, some substrates can serve as allosteric effectors [10,11,12,13,14,15], which can activate or inhibit the metabolism of other substrates [6,10,11,16,17,18]. These effects, termed drug-drug interactions (DDI), represent one of the biggest hurdles on the pathway to the FDA approval of new drugs [19]. While much has been documented about such interactions, it remains a challenge to predict the outcome of these metabolic perturbations for the new drugs in a drug discovery process [20,21,22,23].

In most cases, drug interactions are detected in vitro as the reversible or irreversible inhibition of CYP3A4, and as concomitant diminished clearance of the target substrate. The identification of such effects requires measurements of metabolism at various effector concentrations and the estimation of an IC50. In addition, subsequent in vivo studies may reveal other factors, such as the induction of P450 expression, which may invalidate in vitro conclusions. Understanding the molecular mechanisms of possible DDI mediated by cytochromes P450 would provide a useful conceptual basis for a systematic analysis of the most important factors. Importantly, experimental protocols for in vitro studies should provide quantitative and reproducible measurements of essential biochemical parameters, such as catalytic rates and binding constants. The incorporation of heterologously expressed CYP3A4 in lipid nanodiscs allows one to obtain an array of data supporting the general mechanistic understanding of homotropic and heterotropic cooperativity in CYP3A4 [10,11,24,25,26,27] by defining the inherent properties of states with differing number of molecules bound. Based on previous results obtained with substrates and effectors such as (i) testosterone (TST) and alpha-naphthoflavone (ANF) [27]; (ii) progesterone (PGS) and carbamazepine [11,25]; and (iii) atorvastatin and dronedarone [10], the general picture of homotropic and heterotropic interactions of substrates and effectors in CYP3A4 is available. Using this approach, we outline a simple experimental in vitro approach for the detection and prediction of the possible involvement of new drug candidates in drug-drug interactions mediated by CYP3A4.

CYP3A4 monomer in nanodiscs can bind up to three steroid molecules [11,24], as well as other compounds of similar molecular weight, such as ANF [27] and CBZ [11,25]. Two of these molecules are accommodated inside the substrate-binding pocket, while the third interacts with CYP3A4 at the protein–membrane interface at the pocket formed between the F-F’ and G-G’ loops on one side and lipid head groups on another. This was confirmed by the effect of mutations at the residues 212–215 [25,26], and by the binding of covalently linked dimers of TST at the substrate-binding pocket and the allosteric effect caused by TST monomer [28]. The binding of TST and PGS at this peripheral site with high affinity is also corroborated by two X-ray structures [29,30]. This is schematically shown in Figure 1.

A functional importance of steroid binding at the remote site was shown by multiple spectroscopic and biochemical methods. For example, the binding of TST at this site inhibits the autoxidation of CYP3A4 [32,33], increases the fractions of high spin iron when TST dimer is present at the productive site [28], and improves product yield and coupling in TST or PGS hydroxylation by CYP3A4 [11,24,27]. The binding of PGS at this allosteric site significantly accelerates the epoxidation of CBZ, but the opposite outcome is observed when CBZ as an effector inhibits the hydroxylation of PGS [11].

Similar results were obtained with larger substrates, atorvastatin (ARV) and dronedarone (DND) [10], where the presence of DND at the allosteric site accelerated DND metabolism, while ARV as an effector inhibited product formation from DND. In general, these experimental observations suggest that the sign and magnitude of the effector binding at the allosteric site on the catalytic activity of CYP3A4 depend on the probe substrate and should be tested experimentally.

Overall, most substrates can also appear as allosteric effectors for CYP3A4, when they bind to the peripheral site shown in Figure 1 and perturb the conformational equilibrium and dynamics of F-F’ and G-G’ residues shaping the dome of the substrate-binding pocket. The outcome of such perturbation may be the activation or inhibition of metabolism, depending on the specific positioning and properties of the probe substrate. In addition, the effector can also bind directly at the substrate-binding pocket and inhibit the metabolism of the probe substrate as a competitive inhibitor. Therefore, for each new target compound the evaluation of its potential ability to be involved in significant drug-drug interactions must be tested using a potentially universal experimental protocol, which provides a reference for the measuring and comparing of multiple substances under identical conditions. Such a protocol should distinguish between the direct competitive inhibition of the metabolism of the probe substrate, and various effects (inhibition, activation, or a change in the site of metabolism) caused by interactions with the allosteric site. It should be also relatively simple and easy to scale up to high throughput screening, unlike our own global analysis approach, described above, which provides a detailed mechanistic understanding of heterotropic interactions but requires multiple experiments performed with highly purified structurally homogeneous preparations of CYP3A4 [10,11,24,25,26].

In this work, we propose using midazolam (MDZ) as a probe substrate as a means for testing multiple compounds for potential drug-drug interactions.

Midazolam has been commonly used as a specific substrate for CYP3A4 activity in vitro [3,18,34,35,36,37,38] and in vivo [6,39,40] and for the detection of potent inhibitors of this cytochrome P450 [22,23,41,42,43]. The monitoring of the formation of 1OH-MDZ, the main product of MDZ metabolism, is utilized as a probe for the competitive binding of various drugs to CYP3A4. However, a minor product, 4OH-MDZ, is also generated by CYP3A4, and its fraction reaches ~35% at high MDZ concentrations. Therefore, the metabolism of MDZ is allosterically regulated, and the mechanism of the homotropic cooperative interactions of MDZ has been studied previously by many authors [38,44,45,46,47,48,49]. As described in our recent study [26], we performed the global analysis of the MDZ metabolism by CYP3A4 in nanodiscs and resolved the binding constants and fractions of 1OH and 4OH products formed when one or two MDZ molecules are bound. These results confirmed that midazolam binds inside the active site in a productive position with higher affinity, while a second MDZ molecule interacts with lower affinity with the peripheral allosteric site between the F-F’ and G-G’ loops of CYP3A4 and the membrane lipid head groups. Notably, we found that midazolam is hydroxylated exclusively at the C1 position in the absence of effector bound at the allosteric site, while the presence of the second midazolam molecule or progesterone results in the appearance of ~35% of 4OH product. In addition, in the presence of progesterone, the yield of the minor 4OH product is also favored. This latter fact indicates the existence of heterotropic cooperativity with an allosteric effector at the remote site enhancing the hydroxylation of MDZ at the C4 position.

Based on this information, MDZ can be considered as a sensitive probe for drug-drug interactions mediated by CYP3A4. To extend this understanding, we systematically probed the effects of several steroids and other drugs on MDZ metabolism and on the site of metabolism (SOM) ratio, determined as the ratio of 1OH and 4OH production rates. We found that mutations at residues 211–214 significantly perturb these allosteric interactions, again indicating the involvement of F-F’ loop as the part of CYP3A4 allosteric site in the regulation of the MDZ regiospecificity.

Overall, these results strongly suggest that the 1OH/4OH SOM ratio in MDZ hydroxylation can serve as a sensitive probe for the ability of other drugs to interact with the allosteric site in CYP3A4 and to perturb the metabolism of CYP3A4 substrates. In order to study these interactions in a systematic manner, we first studied these allosteric interactions using MDZ as a substrate and PGS as a well-established effector with known affinity to the allosteric site [26]. Then, we extended these studies to the set of various compounds, including the substrates and effectors for CYP3A4 to compare the extent of the observed allosteric perturbations of the SOM ratio in the MDZ hydroxylation reactions. We found that most of the compounds change the SOM ratio in the same way as MDZ does. However, the effect of ANF and Gefitinib is opposite to that observed with steroids, indicating a different mode of interactions with the allosteric site.

In a previous paper [26], we demonstrated that the SOM ratio of MDZ hydroxylation by CYP3A4 is sensitive to the presence of the second MDZ or PGS as effectors in the allosteric pocket formed by the F-F’ and G-G’ loops. In order to explore the limits of sensitivity of the SOM ratio using MDZ as a probe substrate with various drugs and model compounds as effectors, we initiated a systematic study of the effect of steroids and non-steroid drugs on the MDZ hydroxylation. In these experiments, we use CYP3A4 co-incorporated in nanodiscs with its flavoprotein reductase CPR, although similar metabolic profiles can be obtained with microsomes or other systems.

## 2. Materials and Methods

### 2.1. Compounds

Androstenedione (9001311), corticosterone (16063), medroxyprogesterone (24908), progesterone (15876), testosterone (15645), gefitinib (13166), schisandrin A (19849), α-naphthoflavone (16924), and midazolam (16193) were purchased from Cayman Chemical Company. POPC lipids were purchased from Avanti Polar Lipids. All other chemicals were purchased from Sigma-Aldrich. Structures are shown in the Figure 1.

### 2.2. Protein Expression and Purification

The expression and purification of the membrane scaffold protein (MSP) used for nanodisc formation, cytochrome P450 CYP3A4, and rat P450 reductase, as well as the assembly of CYP3A4 into POPC nanodiscs (ND), were performed following previously described protocols [32,33]. Briefly, cytochrome P450 CYP3A4 was expressed from the NF-14 construct in the PCWori+ vector with a C-terminal pentahistidine tag generously provided by F. P. Guengerich (Vanderbilt University, Nashville, TN, USA). Cytochrome P450 reductase (CPR) was expressed using the rat CPR/pOR262 plasmid, a generous gift from T. D. Porter (University of Kentucky, Lexington, KY, USA). The incorporation of CPR into preformed and purified CYP3A4 nanodiscs was achieved by the direct addition of CPR at a 1:4 CYP3A4:CPR molar ratio, as described previously [50]. All experiments were performed at 37 °C using a POPC Nanodisc system similarly to our earlier mechanistic studies [10,11,24,25,26,27] to allow for the direct comparison of results.

### 2.3. Ultraviolet−Visible Spectroscopy

Substrate titration experiments were performed using 1–2 μM CYP3A4 in nanodiscs with a Cary 300 spectrophotometer (Varian, Lake Forest, CA, USA) at 37 °C. The final concentration of methanol used for substrate solubilization was always <1.5%.

### 2.4. MDZ Hydroxylation, Reactions, and Product Analysis

CYP3A4-incorporated nanodiscs with CPR added at a 1:4 molar ratio in the presence of a substrate were preincubated for 5 min at 37 °C in a 1 mL reaction volume in 100 mM HEPES buffer (pH 7.4) containing 10 mM MgCl2 and 0.1 mM dithiothreitol. The concentration of CYP3A4 was in the range of 60–100 nM. The reaction was initiated with the addition of 200 nmol of NADPH. NADPH consumption was monitored for 5 min and calculated from the absorption changes at 340 nm using an extinction coefficient of 6.22 mM^−1^ cm^−1^.

For product analysis, the reactions were performed in volumes of 80–100 μL in microcentrifuge tubes. At the end of the incubation period, the reactions were quenched by the addition of 20 μL of acetonitrile containing an internal standard (37 μM nordiazepam). The samples were centrifuged at 3000× *g* for 30 min, and 20–40 μL portions of supernatants were injected onto an Ace 3 C18 HPLC column (2.1 mm × 150 mm, MAC-MOD Analytical, Chadds Ford, PA, USA) on a model LC-20LD chromatograph (Shimadzu). The mobile phase contained 22% acetonitrile and 28% methanol in water; products of PGS hydroxylation and MDZ hydroxylation were separated at a flow rate of 0.15 mL/min as follows: isocratic separation for 20 min, then a linear gradient for 5 min during which the concentrations of acetonitrile and methanol were increased to 50% and 32%, respectively, followed by a second isocratic separation for 12 min. The calibration and method validation were performed using commercially available metabolites of PGS and MDZ. The chromatograms were processed with Shimadzu software.

### 2.5. MD Simulation

We prepared conventional all-atom molecular dynamics simulations using the CHARMM-gui software (http://www.charmm-gui.org/ last accessed 15 May 2022) [51,52] as described [26]. Each system consisted of a single full-length CYP3A4 inserted in a lipid bilayer with 360 POPC molecules. The crystal structure of CYP3A4 bound to a single midazolam substrate [53] (protein data bank (PDB) entry 5TE8) was used as a starting structure. The N-terminal helix and other residues missing in the crystal structure were modeled using MODELLER [54]. For the allosteric simulations, a progesterone molecule was docked at the allosteric site using Autodock [55]. We selected a rectangular box between the F-F’ and G-G’ loops as a starting volume for docking. Autodock was then used to find binding configurations of progesterone in that region, and the structure with the lowest energy was chosen as a starting configuration. The system was solvated using TIP3P water molecules and neutralized with 96 sodium ions and 99 chloride ions. The simulation rectangular box was 12.4 nm × 12.5 nm × 13.5 nm in size with periodic boundary conditions. After building the system in CHARMM-gui, we used AmberTools [56] to include the Amber ffSB19 force field for the protein [57], the Amber lipid17 force field for POPC lipids, and the generalized Amber force field [58] to derive parameters for midazolam and progesterone. For the heme ligand, we used the ferric high-spin force field parameters derived in [59]. Parameters for the midazolam and progesterone molecules were generated with the Antechamber software included in AmberTools [56].

The systems were first energy minimized using a steepest descent algorithm if the maximum force did not exceed 1000 kJ mol^−1^ nm^−1^ and next equilibrated in six steps using a Berendsen thermostat while gradually decreasing restraints at T = 310 K. For minimization and equilibration, all bonds to hydrogen atoms were converted to constraints and the LINCS algorithm was employed [56]. For production runs, temperature coupling was performed using a Nosé−Hoover thermostat, and pressure coupling was performed using the semi-isotropic Parrinello−Rahman barostat. The total simulation time of the equilibration was 1.87 ns. After equilibration, we generated eight independent production simulations with MDZ as a substrate and PGD as an effector by randomizing velocities 3.8 μs in total length. These trajectories were compared to the simulations without effector, as described in [26].

## 3. Results

### 3.1. Allosteric Effect of Progesterone

In order to explore the sensitivity of the SOM ratio of MDZ hydroxylation (the ratios of 1OH-MDZ and 4OH-MDZ formation) in the presence of various effectors, we measured the rates of MDZ hydroxylation at different substrate concentrations in the absence and in the presence of various effectors. As reported earlier [26], progesterone binds at the allosteric site and induces significant changes in the SOM ratio (Figure 2). In the presence of PGS, the formation of 4OH-MDZ is favored, and the effect is concentration-dependent. At low MDZ concentrations, the SOM ratio is almost five-fold lower in the presence of 15 µM PGS than in the absence of the effector. At higher MDZ concentrations, this difference is much smaller because MDZ also binds to the same site and induces similar changes in the SOM. For MDZ, the stepwise binding constants were resolved as 5.1 µM (for the productive position at the substrate-binding pocket) and 14.7 µM for binding at the allosteric site [26]. Therefore, the binding of MDZ at the allosteric site is becoming more favorable than PGS at MDZ concentrations higher than 15 µM.

### 3.2. Allosteric Effect of Other Steroids

Next, we tested the sensitivity of the SOM ratio of MDZ hydroxylation in the presence of other steroids, using similar approach. We measured the rates of 1OH-MDZ and 4OH- MDZ production at various MDZ concentrations in the presence of several steroids, namely, androstenedione, corticosterone, testosterone, and medroxyprogesterone, Figure 3.

The concentrations of steroids were 20 µM and 50 µM, except for medroxyprogesterone (2 µM and 8 µM), based on their apparent binding constants to CYP3A4, estimated from the spectral titration curves (data not shown). The binding of medroxyprogesterone was significantly tighter, and lower concentrations of this steroid were used to prevent the complete inhibition of MDZ hydroxylation.

All steroids show the allosteric perturbation of MDZ metabolism, changing the SOM ratio (1OH/4OH). The strongest effect is from testosterone, and the weakest is from corticosterone. For all steroids, this effect is concentration-dependent, which may be the result of the different affinity of these steroids to the allosteric pocket and partly due to possible variations in the interaction mode and in the conformational changes of CYP3A4 caused by these interactions. The allosteric effect of steroids is concentration-dependent, indicating the binding of all the steroids to the remote allosteric site of CYP3A4 because the productive site is occupied by the substrate MDZ. At higher MDZ concentrations, the difference in the SOM caused by the presence of steroids becomes significantly weaker, similar to the tendency observed with PGS, suggesting that MDZ competes with steroids for the allosteric site.

### 3.3. Allosteric Effect of Non-Steroid Compounds

The same set of experiments was performed with several other non-steroid substrates. We used the well-known effector of CYP3A4 ANF [27,44], the non-selective tyrosine kinase inhibitor Gefitinib, used in cancer therapy [60], and the anti-inflammatory natural compound Schisandrin A [61]. These structurally diverse probes were selected in order to study the sensitivity of the MDZ SOM ratio to the presence of effectors with significantly different structures and properties.

In order to probe the possible binding of these drugs in the substrate-binding pocket, we performed the spectral titration of CYP3A4 in nanodiscs with Gefitinib and Schisandrin. The results shown in Figure 4 indicate that spin shift to ~25% was observed with Gefitinib and up to ~70% with Schisandrin, and the titration results could be well described using simple non-cooperative Langmuir isotherm. Based on these results, the concentrations of Gefitinib and Schisandrin were selected in the range between ~0.3 and 1.3 K_S_, in order to avoid the significant inhibition of MDZ hydroxylation, similar to the experimental conditions with steroids as effectors. ANF concentrations were selected in a similar manner based on earlier titration results [27].

The SOM ratios for MDZ hydroxylation measured in the presence of ANF, Gefininib, and Schisandrin are shown in Figure 5. The effect of Schisandrin is similar to that of steroids; it alters the SOM ratio in a similar way but to a lower extent. ANF and gefitinib act differently; in their presence, the SOM changes in an opposite direction, and the hydroxylation of MDZ at C4 position is becoming less favorable. Therefore, the binding of these two effectors at the allosteric site perturbs the conformation and dynamics of F-F’ and G-G’ loops differently, and to a lower extent than steroids and MDZ itself. A similar result with ANF was reported earlier [45].

### 3.4. Molecular Dynamics Simulations

CYP3A4 is highly dynamic, and its substrate-binding pocket undergoes substantial changes in size and shape during MD simulations. The substrate MDZ is also moving with respect to the heme, and rotating so that the orientation of the substrate with respect to the catalytic heme iron–oxygen intermediate changes significantly. Because our goal is to find the most important structural parameters responsible for the site-specific hydroxylation of MDZ, we selected for analysis only those frames in which MDZ was in a “productive” position. We defined the positions of MDZ as potentially productive, when C1 (the main site of metabolism, whose structure is shown in the Figure 1B) is within 4.5 Å of a virtual point corresponding to the oxygen atom of Compound **1**, (FeO) intermediate. This choice of productive position of the substrate is based on the commonly accepted mechanism of P450 catalyzed hydroxylation as hydrogen atom abstraction followed by oxygen rebound. We scanned all eight simulations, 3.8 µs total, for these conformations of CYP3A4 and found 13,879 such structures for simulations without PGS and 14,238 structures with PGS present as an allosteric effector, or slightly more than 50% of all of them. This is consistent with the high uncoupling of MDZ hydroxylation, as measured for CYP3A4 in nanodiscs [26]. In all simulations with PGS with one exception, PGS stays in contact with the F-F’ loop and with lipid headgroups. In the end of one simulation, PGS dissociated and moved away to the solvent.

We analyzed all the productive structures for the relative distances of C1 (major site of metabolism) and C4 (minor site) from the virtual point corresponding to the oxygen atom of Cpd **1** and divided them in two groups: selective and non-selective. The selective MDZ orientations correspond to those where C4 is significantly further away from heme and cannot be effectively hydroxylated. Non-selective are those where C4 is also not far from the heme and is in a position favorable for hydroxylation by Cpd **1**. A comparison of these two groups for MD simulations in the presence and in the absence of PGS as an allosteric effector is shown in Figure 6.

Figure 6 shows that the probability of C4 moving closer to the heme iron increases when PGS is present at the allosteric site, while the fraction of conformations with C4 moving away from productive position (more than 5.5 Å) significantly decreases. This indicates the presence of allosteric effect results from PGS binding at the peripheral pocket between F-F’ and G-G’ loops on the SOM ratio of MDZ hydroxylation, similar to the one observed in the experiment and to the same results obtained earlier for MDZ homotropic cooperativity.

The statistical analysis of the number of close contacts of CYP3A4 residues with substrate MDZ is shown in Figure 7. Frames selected as productive and summarized in Figure 6 were analyzed for the residues, which directly interact with MDZ and define its position and orientation. The differences between simulations in the absence and in the presence of PGS at the allosteric site allow one to identify the residues responsible for allosteric signaling, resulting in the less restricted positioning of MDZ and a significant increase of production of minor metabolite 4OH-MDZ.

In these simulations, the binding of PGS as an effector is seen to induce the significant loss of MDZ contacts with residues 106 and 108 in the B-C loop and residues 215, 218, and 220 in the F-F’ loop. In addition, contacts with residues 217 and 309 are also becoming rarer, but these numbers are relatively lower, i.e., they represent a decrease of less than 50%. On the other hand, contacts of MDZ with residue 214 occur significantly more often in the presence of PGS, and residues 304 and several others are also more often in contact with MDZ, although these increases are significantly less than 50%. Overall, these observations are consistent with experimentally observed the higher mobility of the substrate MDZ in the presence of effector PGS, as the total number of lost contacts with CYP3A4 residues decreases as a result of allosteric effector binding.

A representative example of the conformational changes observed in the presence of PGS at the allosteric site is shown in Figure 8. Two frames from MD simulations without PGS (selective position of MDZ at the active site is shown) and in the presence of PGS corresponding to non-selective position of MDZ (not shown) are superimposed with the heme plane used as a reference. The outward movement of the residues 215, 217, and 218 expands the volume of the substrate-binding pocket and favors the mobility of the MDZ molecule, concomitant to the increase in the minor product formation. This movement is happening because residues F213, F220, and L221 are interacting with the effector PGS, and these interactions cause the outward movement of the neighboring residues at the F-F’ loop. A similar effect was observed in our earlier work [26], where the binding of PGS at the allosteric site caused the outward movement of F213 and F215, which in turn improved the packing of carbamazepine substrate and accelerated its metabolism.

## 4. Discussion

Understanding the mechanism of allosteric interactions in CYP3A4 is important for the development of new drugs and for the prediction of drug-drug interactions. Although a complete picture of CYP3A4 allostery may not yet be available, the essential details schematically shown in Figure 1 can be summarized as follows. The binding of effector molecules at the peripheral allosteric site formed between F-F’ and G-G’ loops of CYP3A4 and the lipid head groups of the membrane changes the conformation and dynamics of these loops. At the same time, the residues 212–220 and 231–240 belonging to these loops constitute the part of the substrate-binding pocket and are in direct contact with substrates bound at the active site. Therefore, conformational changes of these residues caused by the presence of the effector at the allosteric site may favor a change in position of substrates in the active site and, consequently, the rates and site of metabolism. The result of such perturbations caused by any given effector can be different for different substrates, i.e., the metabolism of some substrates can be inhibited, but for others it can be accelerated. Such an example was observed for the DND-ARVS pair, where DND as an effector accelerates the metabolism of DND as a substrate but inhibits the metabolism of atorvastatin [10]. Allosteric activation was also reported for other human cytochromes P450 [62], including cholesterol hydroxylation by CYP46A1 in vivo [63,64] and CYP2J2 [65,66].

The results obtained in this study, together with earlier publications, provide an experimental basis for simple and sensitive screening for potential heterotropic drug-drug interactions mediated by CYP3A4. Using MDZ as a probe substrate, allosteric interactions with the peripheral site at the protein–lipid interface can be detected by monitoring the SOM ratio, i.e., the ratio between the major C1-hydroxylated product and the minor C4-hydroxylated product. The effect of various compounds is concentration-dependent, indicating their competitive displacement of the probe substrate MDZ, which also acts as an effector. The latter observation provides a useful measure of the relative affinity of tested compound binding to the allosteric site. This method of detection of allosteric effectors for CYP3A4 as an alternative source of potential drug-drug interactions complements the commonly accepted monitoring of drug interactions caused by competitive inhibition, where the rate of the main product of MDZ hydroxylation is measured in the presence of alternative substrates or inhibitors.

The position of the substrate and SOM preferences may be changed by mutations [67,68,69] and by interactions with allosteric effectors [6,26,44,70]. Vice versa, concentration-dependent changes in the SOM ratio due to the presence of various compounds can be used as a direct indication of ability of these compounds to serve as allosteric effectors. In many cases, these effectors may also bind inside a substrate-binding pocket and thus serve as alternative substrates and/or competitive inhibitors of the metabolism of the target substrate. An analysis of the SOM ratio allows one to distinguish between these two mechanisms of drug-drug interactions and to detect effectors that can interact with the peripheral allosteric site and thus may perturb the metabolism of the substrate without entering the active site of CYP3A4. In some cases, the binding of the effector at the peripheral site can change the rate of the substrate metabolism [4,10,11,26].

## Data Availability

The data used to support the findings of this research are available upon request.

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
