# Peer review of "Midazolam as a Probe for Heterotropic Drug-Drug Interactions Mediated by CYP3A4"

_biomolecules, 2022, doi:10.3390/biom12060853_

Round 1
Reviewer 1 Report
In this work I.G. Denisov et al. developed a method for studying the potential involvement of drug-drug interactions in CYP3A4. The site of metabolism ratio is used to estimate the allosteric heterotropic interactions caused by allosteric small molecules effectors. Moreover, molecular dynamics simulations were performed to rationalize the experimental results. This has shown that conformational changes in the F-F’ loop could be related to the modified catalytic activity of the enzyme.
Generally, I find this manuscript interesting and sounding, in my opinion this work can be more suitable for Biomolecules after some minor revisions are performed. Here is a list of my comments/concerns:
- Figure 1: Details in the description are missing, e.g blue surface: active site? Red heme. Maybe would be clearer if the position of the N-term helix crossing the membrane is also shown in the figure.
- Line 254: Paragraph 3.1 is empty. Check it.
- Line 324: Show in a clearer way C1 and C4 position in MDZ. Maybe add an inset in Figure 6 or modify Scheme1.
- Figure 6: Units are missing in the probability axes.
- Line 356: Definition of close contact is missing here (e.g. cutoff used?)
- I find Figure 7 confusing, especially the overlapping histograms in the left panel. The authors should try to make it clearer.
- Line 368: are some of the residues listed here known in the literature? (e.g. mutagenesis studies, key catalytic residues). Please expand.
- Figure 8: How these frames were selected? Maybe it would have been better to consider frames obtained from a cluster analysis of the MD trajectories.
- Line 391: This conformational change is interesting, and I think should deserve a deeper investigation (e.g. using principal component analysis or similar methods).
Author Response
Reviewer #1. Specific comments, addressed point by point:
Figure 1: Details in the description are missing, e.g blue surface: active site? Red heme. Maybe would be clearer if the position of the N-term helix crossing the membrane is also shown in the figure.
We fixed the legend, As for N-terminal helix, it is at the back of the picture and cannot be seen well
- Line 254: Paragraph 3.1 is empty. Check it.
We corrected this error and added the following paragraph.
“Binding of PGS at the allosteric site significantly decreases the 1OH/4OH ratio of MDZ hydroxylation by CYP3A4 in Nanodiscs. At the constant concentration of PGS which was kept at 15 uM, the heterotropic effect is becoming less pronounced with the increase of MDZ concentration because of the competition between MDZ and PGS for the allosteric site. For MDZ the stepwise binding constants were resolved as 5.1 uM (for the productive position at the substrate binding pocket) and 14.7 uM for binding at the allosteric site [25] . Therefore, binding of MDZ at the allosteric site is becoming more favorable than PGS at MDZ concentrations higher than 15 uM.”
- Line 324: Show in a clearer way C1 and C4 position in MDZ. Maybe add an inset in Figure 6 or modify Scheme1.
Scheme 1 shows positions of C1-hydroxylated and C4-hydroxylated products, we added a reference in the text
- Figure 6: Units are missing in the probability axes.
We added units to Figure 6
- Line 356: Definition of close contact is missing here (e.g. cutoff used?)
We used 4.5 angstroms as cutoff value, now added to the text.
- I find Figure 7 confusing, especially the overlapping histograms in the left panel. The authors should try to make it clearer.
We replaced this figure with a new Figure 7 to avoid overlapping and provide more clear illustration of conformational changes observed in MD simulations.
- Line 368: are some of the residues listed here known in the literature? (e.g. mutagenesis studies, key catalytic residues). Please expand.
Various residues at the F-F’ loop have been reported to play a role in the allosteric interactions and cooperativity in CYP3A4, however, to our knowledge, no specific role of any of the residues listed in the text under Figure 8 was suggested for alteration of MDZ hydroxylation. This may become a subject for future studies.
- Figure 8: How these frames were selected? Maybe it would have been better to consider frames obtained from a cluster analysis of the MD trajectories.
The selection was based on comparison of distances of several residues identified at Fig. 7 from the heme plane, illustrating the movement of the F-F’ loop outward from the substrate, which was caused by PGS binding
- Line 391: This conformational change is interesting, and I think should deserve a deeper investigation (e.g. using principal component analysis or similar methods).
We agree that this conformational change deserves more attention, especially in the light of our earlier results (ref. [11]). It may become the subject of further MD studies of allosteric site of CYP3A4.
Reviewer 2 Report
This work is an installment in a series of nice papers from this lab that tease out the detailed structural and dynamic effects of drugs on CYP3A4 that could lead to lead to allosteric drug-drug interactions not caused by the simpler competitive inhibition. In this paper the authors employ additional MD simulations to understand the effects of some steroids and other drugs on the metabolic ratio of products obtained from midazolam. The work is interesting and valuable at a fundamental level.
My major concern with the work is the naivete, or maybe the conscious overstatement, of the suggestion that monitoring the product ratio of midazolam has any practical utility in predicting DDIs and that it is a ‘new’ approach. The product ratio for MDZ has been appreciated for a long time and used by many others to probe functional properties and drug interactions with CYP3A4. There is a vast literature from other labs, possibly under cited by the authors, that describes the MDZ product ratio and even its sensitivity to other drugs, including the application of MD simulations by Hackett et al. No drug development pipeline is ever going to use CYP3A4 in nanodiscs and monitor the MDZ product ratio as a screen for DDIs. Everyone already uses panels of CYPs with isoform-specific probes with well-established isoform-dependent perpetrator drugs compared to drug candidates, using a readout of decrease/change in rate of metabolite formation to assess the potential for an increase in AUC of candidate drug (victim) in the presence of known perpetrators and vice versa. Allosteric drug interactions are known to be possible in vivo, in principle, based on the same product ratio of 1-OH vs. 4-OH MDZ, as referenced by the authors (ref. 7). However, allosteric interactions are quantitatively not a significant contribution to AUC-based DDI’s as far as anyone knows. The authors approach can detect interactions between other drugs and MDZ, as already shown by others, and it adds to the understanding of how some additional drugs interact with MDZ. However, it doesn’t provide a useful new method for detection of in vitro DDIs.
Perhaps most importantly, the authors even seem to appreciate that the effect of drugs on MDZ product ratio is context dependent, or ‘perpetrator’ dependent. In turn, DDIs are both victim dependent and perpetrator dependent. So, the effect of any new drug on the product ratio of metabolites from MDZ as a ‘victim’ is unlikely to be a great predictor of DDI potential by the new drug with other, non MDZ, drugs. It is just unrealistic to think a drug company would go to the time and expense to create CYP3A4 nanodiscs to look at MDZ metabolism instead of looking at panels of drugs in microsomes. The authors should recast the value of their experiments in a more fundamental level. Their work nicely explains some structural and dynamic effects that are relevant for the specific case of MDZ as a ‘victim’.
That said, the authors also possibly understate the well appreciated importance of the F-F’ region in mediating allosteric effects. This is pretty well traveled territory. In fact as far back as 1998 Harlow and Halpert made mutations at residues 211 and 214 and documented their role in allosteric binding (PNAS 95:6636). Yes, each specific drug or combination of drugs has slightly different effects on the local dynamics around the Phe-cluster or the F-F’ loop, as noted for two MDZ’s bound or multiple Testosterones bound in other publications (Hackett JBC 293:4037; Hackett et al Biochemistry 59:766). So, this work adds additional detail about drug-specific effects but the role of this site on drug metabolism, including MDZ, is established. The paper adequately addresses many papers that deal with CYP3A4 allostery but among those it seems to ignore a couple most closely related to this work. This work adds some detail to a well studied behavior but it doesn’t really provide a ‘new approach.’
Specific questions:
Figure 7. The figure needs y-axes labels.
Figure 6 and associated discussion. It seems as if the authors have sampled different poses of MDZ and enumerated the relative frequency (probability) that the 4-Carbon of MDZ sits at various distance (selective) from the ‘virtual’ iron-oxo species. The PGS shifts the frequency to shorter distances. However, don’t we need to know what happens to C1 in order to determine whether the SOM will change – is it possible that both C1 and C4 move closer without affecting the SOM? Is the C4 distance by itself an appropriate marker for the SOM?
Author Response
Reviewer #2. Specific comments, addressed point by point:
Figure 7. The figure needs y-axes labels.
We made a new Figure 7 and added Y-axis labels as noted above.
Figure 6 and associated discussion. It seems as if the authors have sampled different poses of MDZ and enumerated the relative frequency (probability) that the 4-Carbon of MDZ sits at various distance (selective) from the ‘virtual’ iron-oxo species. The PGS shifts the frequency to shorter distances. However, don’t we need to know what happens to C1 in order to determine whether the SOM will change – is it possible that both C1 and C4 move closer without affecting the SOM? Is the C4 distance by itself an appropriate marker for the SOM?
We calculated the distribution of distances from C4 to the iron-oxo catalytic site as the surrogate for the productive positioning with respect to MDZ hydroxylation at C4 position. Only the frames with productive C1 positioning are samples for this figure. The number of ‘unproductive’ positions for C1 while C4 is productive, is negligible comparing to ~14000 frames analyzed. As for the SOM ratio dependence on the C1 and C4 distances addressed in reviewer’s question, we do not have any quantitative method to estimate this dependence. The cutoff distances 5 angstroms from oxygen atom to the C1 or C4 atoms were selected as the criteria for the possible hydrogen abstraction from MDZ molecule.
R2 general criticism
My major concern with the work is the naivete, or maybe the conscious overstatement, of the suggestion that monitoring the product ratio of midazolam has any practical utility in predicting DDIs and that it is a ‘new’ approach. The product ratio for MDZ has been appreciated for a long time and used by many others to probe functional properties and drug interactions with CYP3A4.
We agree that the product ratio for MDZ is known as a good indicator of homotropic and heterotropic allosteric interactions in CYP3A4, and we tried to provide a representative overview of earlier studies in the introduction. However, the examples of using this product ratio as a probe of heterotropic drug interactions with CYP3A4 are rare, in almost all cases the measurements are limited to the rate of the main 1OH-MDZ production, used as a good probe for competitive inhibition of CYP3A4 by target compounds. And most articles which report the changes of the product ratio in the presence of other CYP3A4 substrates do not provide concentration dependent effects and do not suggest to use this ratio as a functional probe with other potential substrates or effectors of CYP3A4, which behavior is unknown.
There is a vast literature from other labs, possibly under cited by the authors, that describes the MDZ product ratio and even its sensitivity to other drugs, including the application of MD simulations by Hackett et al. No drug development pipeline is ever going to use CYP3A4 in nanodiscs and monitor the MDZ product ratio as a screen for DDIs. Everyone already uses panels of CYPs with isoform-specific probes with well-established isoform-dependent perpetrator drugs compared to drug candidates, using a readout of decrease/change in rate of metabolite formation to assess the potential for an increase in AUC of candidate drug (victim) in the presence of known perpetrators and vice versa.
These statements are correct, and monitoring of the rate of hydroxymidazolam (almost always only 1OH-MDZ, the main product) is commonly accepted in screening for DDI. Our work provides more insight into the different mechanism of allosteric interactions between substrate MDZ and various effectors, which does not involve a readout of changes in rate of metabolite formation. Alternatively, we suggest that monitoring the site of metabolism ratio, or product ratio, provides a sensitive and simple probe for obtaining additional information on the possible involvement of new drug candidates in the DDI mediated by CYP3A4.
We do not suggest using nanodiscs for this type of screening, the product ratio with MDZ as a substrate can be measured in multiple in vitro systems, as well as in living organisms. This is why we think it is important to attract attention to this approach in DDI studies. We do not claim that monitoring this site of metabolism ratio is discovered by us, however we trust that our work contributes new and useful information for more common application of this method.
Allosteric drug interactions are known to be possible in vivo, in principle, based on the same product ratio of 1-OH vs. 4-OH MDZ, as referenced by the authors (ref. 7). However, allosteric interactions are quantitatively not a significant contribution to AUC based DDI’s as far as anyone knows. The authors approach can detect interactions between other drugs and MDZ, as already shown by others, and it adds to the understanding of how some additional drugs interact with MDZ. However, it doesn’t provide a useful new method for detection of in vitro DDIs.
This approach can detect interactions between other drugs and MDZ, as confirmed by this reviewer. Based on this statement, we trust that measuring the product ratio of MDZ hydroxylation in the presence of different concentrations of new drug candidates provides useful information on the ability of these molecules to bind to the effector site and to cause changes in conformation and dynamics of the active site of CYP3A4. Sometimes such allosteric changes can cause activation or inhibition of substrate metabolism, as it was shown for CYP3A4, CYP46A1 and other human cytochromes P450, as we mentioned in our manuscript (references [4, 10, 11, 62-66]). We think that the product ratio probe with MDZ is an efficient and useful test for such interactions, and it can be used with microsomes, hepatocytes, baculosomes, or reconstituted purified CYP3A4, not only CYP3A4 in nanodiscs.
Perhaps most importantly, the authors even seem to appreciate that the effect of drugs on MDZ product ratio is context dependent, or ‘perpetrator’ dependent. In turn, DDIs are both victim dependent and perpetrator dependent. So, the effect of any new drug on the product ratio of metabolites from MDZ as a ‘victim’ is unlikely to be a great predictor of DDI potential by the new drug with other, non MDZ, drugs.
We agree with reviewer that the results of such changes depend on the specific properties of both “perpetrator’ (effector) and substrate (‘victim’), but this fact does not preclude one from assessing the possible involvement of new drug candidates in such interactions by using one or several standard substrates as reference probes, providing a necessary base for comparison of “perpetrator” potential.
It is just unrealistic to think a drug company would go to the time and expense to create CYP3A4 nanodiscs to look at MDZ metabolism instead of looking at panels of drugs in microsomes. The authors should recast the value of their experiments in a more fundamental level.
The product ratio (site of metabolism ratio) for MDZ hydroxylation does not require application of nanodiscs and can be performed using any reliable in vitro system (microsomes, hepatocytes, baculosomes, reconstituted purified CYP3A4, etc.) or in vivo, depending on the specific requirements at the given lab, be that academia or pharmaceutical company. We have added a comment to address this issue in the discussion, l. 433-435
Their work nicely explains some structural and dynamic effects that are relevant for the specific case of MDZ as a ‘victim’. That said, the authors also possibly understate the well appreciated importance of the F-F’ region in mediating allosteric effects. This is pretty well traveled territory. In fact as far back as 1998 Harlow and Halpert made mutations at residues 211 and 214 and documented their role in allosteric binding (PNAS 95:6636). Yes, each specific drug or combination of drugs has slightly different effects on the local dynamics around the Phe-cluster or the F-F’ loop, as noted for two MDZ’s bound or multiple Testosterones bound in other publications (Hackett JBC 293:4037; Hackett et al Biochemistry 59:766). So, this work adds additional detail about drug-specific effects but the role of this site on drug metabolism, including MDZ, is established. The paper adequately addresses many papers that deal with CYP3A4 allostery but among those it seems to ignore a couple most closely related to this work. This work adds some detail to a well studied behavior but it doesn’t really provide a ‘new approach.’
We added references and extended the review of the published works on MDZ metabolism, including suggested reference on the recent paper from Hackett and Atkins groups (ref. [46]). The critical importance of the F-F’ loop in allosteric mechanism of CYP3A4 is known, and we tried to cite many important articles on that subject. However, we think that our manuscript indeed provides new information on the various types of concentration dependent effects caused by different chemical compounds (i.e. steroids vs. gefitinib, schisandrine and ANF). These data support the possibility to probe for the interactions of new drug targets with the allosteric site and to detect the possible perturbations (or the lack of those) of the substrate binding pocket caused by these interactions. If necessary, this method can be extended by using other probe substrates instead of MDZ.